# Psychometric Properties of the Knowledge of Hydration among Foreign Students of Óbuda University, Hungary

**DOI:** 10.3390/healthcare12111152

**Published:** 2024-06-06

**Authors:** Melvin Omone Ogbolu, Olanrewaju D. Eniade, Miklós Vincze, Miklós Kozlovszky

**Affiliations:** 1BioTech Research Center, University Research and Innovation Center, Óbuda University, Bécsi Str. 96/b, 1034 Budapest, Hungary; miklos.vincze@uni-obuda.hu; 2Department of Epidemiology and Medical Statistics, College of Medicine, University of Ibadan, CW22+H4W, Queen Elizabeth II Road, Agodi, Ibadan 200285, Nigeria; olanrewaju.eniade@ifain.org; 3International Foundation against Infectious Disease in Nigeria (IFAIN), 6A, Dutse Street, War College Estate, Gwarimpa, Abuja 900108, Nigeria; 4John von Neumann Faculty of Informatics, Óbuda University, Bécsi Str. 96/b, 1034 Budapest, Hungary; kozlovszky.miklos@nik.uni-obuda.hu; 5Medical Device Research Group, LPDS, Institute for Computer Science and Control (SZTAKI), Hungarian Research Network (HUN-REN), Kende Str. 13-17, 1111 Budapest, Hungary

**Keywords:** adequate water intake (AWI), health, hydration, knowledge, knowledge of hydration (KH-11) tool, total water intake (TWI), water

## Abstract

It is known that the quantity, makeup, and distribution of bodily fluids have a significant impact on the cognitive health, physiological health, and cell activity of human beings. This narrative could be influenced by the level of knowledge about hydration, dehydration, and the practice of Adequate Water Intake (AWI) of an individual based on the recommended daily Total Water Intake (TWI) by either the World Health Organization (WHO) or the European Food Safety Authority (EFSA). In this study, we have developed and validated a scale to adequately measure knowledge of the recommended daily Total Water Intake (TWI) practices among foreign students at Óbuda University, Hungary. Hence, we implemented an 11-item scale to measure the Knowledge of Hydration (KH-11) and evaluate its psychometric properties among students. This study is an online cross-sectional study assessing water intake knowledge with the use of the KH-11 tool among 323 students with ages ranging from 18 to 35 years, who have enrolled for at least two semesters at the University. The statistical analysis performed was reliability (using Cronbach alpha ≥ 70%) and factor analysis. Knowledge levels were categorized as poor (<50%), intermediate (50–70%), or adequate (71–100%). The intraclass correlation, chi-square, and rotated component matrix were also estimated and reported. Data were analyzed using SPSS version 25. Cronbach’s alpha analysis revealed that the KH-11 had an overall good reliability with a value of 0.80, where the survey items had an acceptable level of consistency ranging from 0.75 to 0.81 and demonstrated sufficient independence from each other as Pearson’s *R* within factors was positive and ranged from 0.02 to 0.74. In evaluating the participants’ knowledge of hydration, the total possible score for the scale is 72, while the mean score for the KH-11 was 55.2 ± 11.61 SD, and the factor analysis model yielded an acceptable fit (*χ*^2^ = 3259.4, *p* = 0.000). We recorded a high-level positive concordance of 0.770 with an average intraclass correlation of 0.80 at a 95% CI, where *p*-value = 0.000. Our findings show that the majority (66.3%) of the students have a good knowledge of hydration. However, the skewed distribution of the knowledge scores suggests that some may have lower levels of knowledge, which may warrant further study to improve knowledge in those students.

## 1. Introduction

Maintaining fluid and electrolyte balance in living organisms is known as homeostasis, which is crucial for the healthy functioning of the human body [1]. Salt and water overload, hypervolemia, and dehydration have all been linked to adverse outcomes for the human body and how it stays healthy [2]. The water balance of the human body is maintained by regulating its intake (by adopting Adequate Water Intake (AWI) and Total Water Intake (TWI) [3,4]) with water loss; thus, it is an essential component of life [3]. Water is a component of all living cells and extracellular fluids [5]; thus, epidemiological studies have since revealed that humans can only survive for a few days without drinking water [6,7,8]. Hence, it is considered as one of the most vital nutrients required for cellular homeostasis and life.

Water makes up between 60 and 70% of an adult’s body weight, thereby contributing to the regular functioning of the human organs (and the entire human body) [9,10]. It also serves as a solvent, releasing a variety of solutes for use by cells, and it is the medium required for all processes [11]. The physiological functions of digestion, absorption, and elimination all require water to function through a conscious practice of the recommended TWI [6]. A few years ago, the World Health Organization (WHO) agreed with the European Food Safety Authority (EFSA) concerning the relevance of water to human health with regards to the recommended daily TWI with a similar measurement of total fluids intake in Liters (L). According to the WHO, daily TWI is 2.9 L for an adult male and 2.2 L for an adult female who lives a sedentary lifestyle, while for those who live an active lifestyle, it is 4.5 L for an adult male and 4.5 L for an adult female [12]. While for the EFSA, the daily TWI for an adult female is 1.6 L, and the adult male can take 2.0 L [13]. It is said that if any of these recommendations are not adopted, then the human body will suffer from water deprivation [14].

Water deprivation occurs when the balance between daily TWI and loss of body water is disrupted, thus causing a state of dehydration in the human body [15] which may lead to mild/chronic illness [5]. Dehydration can be defined as a decrease in daily TWI content due to fluid loss, reduction in fluid intake, or even both [16]. Dehydration could occur if a person loses as little as 3% of their body weight due to water loss. However, water loss of more than 5% of the total body water would probably lead to mild dehydration, while severe dehydration occurs when there is a 9–10% loss of the total body water [3]. The lack of adequate daily TWI has been revealed to contribute to body weight loss, deteriorating memory, and reduced levels of concentration. Many research studies have found specific medical evidence that shows the association between dehydration (as a result of inadequate water intake and some leading risk factors such as lack of physical effort, exposure to harsh climatic conditions (i.e., heat and humidity), inadequate water intake, and other factors connected to lifestyle [17]) and some chronic diseases or disorders [18]. Just to mention a few of these chronic diseases and disorders, they are kidney stones, heart disease, stroke, cystitis, bowel and colon cancer, stomach ulcers, constipation, excess weight and obesity, diarrheal, digestive disorders, high and low blood pressure, acid–alkaline imbalance, premature aging, skin disorders (e.g., eczema), and energy loss (e.g., fatigue, dizziness, restlessness, etc.) [3,19].

The WHO guideline for TWI is based on a comprehensive review of scientific literature, including studies on water balance, hydration status, and health outcomes. It takes into consideration recommendations from other reputable sources, such as the National Academies of Sciences, Engineering, and Medicine and the European Food Safety Authority. Even though TWI is said to vary by gender, individual water requirements may also vary depending on factors such as climate, physical activity level, and health status [20].

## 2. Study Motivation

According to existing epidemiological studies, public awareness of hydration and dehydration is not widely assessed, and oftentimes, the result reveals that students do not have adequate knowledge about hydration and dehydration because there is a gap in knowledge concerning this subject [21]. Hence, the first phase of this research is to organize seminars within Óbuda University in order to collect adequate data from the students to evaluate their level of knowledge about hydration, while the second phase of this project will examine the efficiency of our survey and Hydration Campaign in proving knowledge of hydration among the students, considering that seminars and campaigns are a means to improve knowledge concerning any topic. To justify that there is a lack of knowledge of hydration and dehydration among students, a study conducted a hydration assessment in 2017 among university students in Southern Nigeria; the authors reported a 46.4% dehydration rate while 59% had inadequate water intake. The study further revealed that the likelihood of coffee intake for dehydration was higher among the study participants [22]. Another study conducted in 2011 among some Chinese scholars also revealed the knowledge of dehydration during summer in four cities, and they reported that only 28.4% knew about dehydration [23]. In 2017, a study assessed dehydration status among university club athletes in Europe and found that dehydration was common (40%) among Karateka, female netball players, army officer cadets, and golfers [24]. Furthermore, a systematic review has reported that there is a major gap in knowledge and measurement of fluid intake and hydration status [9]. Recently, a link has been established between knowledge and healthy behavior, implying that the knowledge of people on daily TWI will influence the practice of AWI [25,26]. Therefore, to understand the knowledge of water intake among any population, the development of scales to adequately measure water intake is required. Although some studies have attempted to assess knowledge of dehydration among diverse populations, there is a lack of clarity on the validity and reliability of the measurement of dehydration. However, after a holistic review, there is a paucity of information on scale development for the measurement of water intake among foreign university students in Hungary. Hence, this study aims to develop a scale (Knowledge of Hydration 11 items (KH-11)) and evaluate its psychometric properties among foreign students in Óbuda University, Hungary, on the practice of TWI as recommended by the WHO or EFSA.

## 3. Methods

### 3.1. Study Design

An online cross-sectional study was conducted to assess the knowledge and practice of water intake among the foreign students of John von Neumann Faculty of Informatics, Óbuda University, Hungary, in October 2019 [3]. The study population was international (foreign) students at Óbuda University. A purposeful sampling method was used due to ease of accessibility and contextual understanding of the study population. The survey was targeted towards only the students of John von Neumann Faculty of Informatics, and the link was shared through emails, the University’s social media platform (Facebook—University and University’s dormitory Facebook page), and the University’s website.

### 3.2. Sample Size

After the distribution of the survey for a period of 3 months, a total of 395 foreign students filled out the survey, but not all participants were included in this study after data processing. However, only 323 foreign students were included in this study. The sample size formula for a cross-sectional study was used, which resulted in a sample size of 323, with the standard normal deviation of 95% confidence level (CI); thus, (Zα) = 1.96, *p* = 0.7, and level of precision = 5% for knowledge of hydration as identified in a previous study [3].

### 3.3. Inclusion and Exclusion Criteria


*Inclusion Criteria are as follows:*


i.Students aged 18–35 years;ii.Males and females only;iii.Foreign students of Óbuda University, Hungary (enrolled for at least two semesters).


*Exclusion Criteria are as follows:*


i.Students under 18 years or above 35 years;ii.Students who indicated not belonging to African, Asian, and European continents;iii.Non-gendered students.

### 3.4. Instrument—Knowledge of Hydration 11 Item Scale (KH-11)

The KH-11 scale (11 items) was developed for the assessment of knowledge of water intake among the students. The questionnaire design was led by the first author, and variables included in the questionnaire were obtained through a literature review by the first and second authors, as well as further consultation with a health economic expert within the institution. The first page of the questionnaire contained the informed consent, which consisted of information about the study, voluntary participation, and the ability to withdraw consent at any point. The study contained 21 questions (which included questions to capture the participants’ demographic characteristics, knowledge, and perception about hydration and dehydration status). In this study, we have extracted 11 knowledge-based questions on hydration to estimate the validity. The scale consists of the following 3 subscales: 1. Importance of water intake for staying hydrated [one item]; 2. Recommendation of Total Water Intake (TWI) and source (Personal recommendation and recommendation according to the European Food Safety (EFSA) and the World Health Organization (WHO) [four items]); 3. Knowledge of the benefits of Adequate Water Intake (AWI) [six items].

The information in Table 1 below captured the questions for each item, options, the scores, and the scoring guide.

According to the table above, the sum of all the items resulted in a total of 80 possible scores. The KH-11 scale has good reliability having a Cronbach’s alpha value of 80%. The subscale had an acceptable internal consistency as Cronbach’s alpha for the subscales ranged from 74.5% to 80.8%.

### 3.5. Data Collection

An electronic version of the survey, which comprises the participants’ demographics, knowledge of the hydration scale, and other related questions, was set up on the Survey Planet survey website. The survey was administered by sending the survey link to the study population through Óbuda University’s social media and official platforms. The data collected includes mostly quantitative data, where we implemented multiple choice and ranking scales as our question types in the survey.

### 3.6. Data Analysis

The data were analyzed using IBM SPSS Statistics version 26. Descriptive statistics were analyzed and presented in frequencies and percentages. We determined the internal consistency and reliability of the item scales and the subscales of the survey using Cronbach’s alpha. The Cronbach alpha acceptability threshold of 70% or higher was adopted for this study [27]. Also, intraclass correlation and chi-square were estimated and presented. Factor analysis was performed to examine the structure of the factors for the KH-11 tool, and the rotated component matrix was presented. We categorized the knowledge score into three groups (<50% score = poor knowledge, 50–70% = intermediate knowledge, and 71–100% = adequate knowledge). This is to be able to analyze and present the level of knowledge of hydration among the study participants.

### 3.7. Ethical Consideration

The fore page of the electronic survey presented adequate information about the study, and participants were informed that participation in the study was voluntary and that no experiment needed to be performed. Hence, ethical approval for this research was not required. All tenets of Helsinki’s declaration were ensured in this study. The data was downloaded on a personal computer, which is password-protected and only accessible to the researchers (i.e., the authors of this study).

## 4. Results

### 4.1. Demographic Characteristics of the Study Participants

The results in Table 2 below present the demographic characteristics of the study participants. Of the participants, 169 (51.7%) were aged 18 to 24 years, while 48.3% of them were aged 25 to 35 years. Similarly, over half of 182 (56.4%) of the participants were males and 141 (43.6%) were females. The majority, 118 (36.5%) of the students were Europeans, 104 (32.1%) were from Africa, and 101 (31.3%) were Asians. All 323 (100.0%) of the participants included in this study resided in Hungary when the study was conducted.

### 4.2. Distribution, Characteristics, Internal Consistency, Reliability, and Homogeneity of the KH-11

The distribution and characteristics of the KH-11 tool are presented in Table 3 below (and Appendix A). The KH-11 scores range from 0 (very low knowledge) to 72 (very high knowledge). The mean score was revealed to be 55.2 ± 11.61. We observed that the distribution of the knowledge score is skewed to the left. Also, the results showed that the KH-11 had an overall good reliability with a Cronbach’s alpha value of 0.80. The items had an acceptable level of consistency with the item Cronbach’s alpha value ranging from 0.75 to 0.81. All the items in the survey demonstrated sufficient independence from each other as Pearson’s correlation (R) within factors was positive and ranged from 0.02 to 0.74.

### 4.3. Participants’ Knowledge about Hydration

As presented in Table 4 below, the total score for the scale is 72, and the mean score for the KH-11 tool is estimated as 55.2 ± 11.61 SD. Also, the mean score for Subscale 1. (Importance of water intake for staying hydrated [one item]) was 0.9 ± 0.07. Subscale 2. (Recommendation of Total Water Intake (TWI) [four items]) had a mean score of 6.3 ± 3.60, and Subscale 3. (Knowledge of the benefits of Total Water Intake (TWI) [six items]) had a mean score of 47.9 ± 11.17.

### 4.4. Factor Analysis Results

The results in Table 5 below show the factor loading for the KH-11 tool. The results also revealed an excellent factor loading for the subscales as well. For instance, all items (six items) measure the knowledge of the benefits of Total Water Intake (TWI) loaded on factor 1 with a factor loading > 0.5. Similarly, for four items loaded on the second subscale (recommendation of Total Water Intake (TWI) and source), the factor loading was >0.9. Only one item was loaded on the third subscale (importance of water intake for staying hydrated) with a factor loading >0.9. Overall, the model yielded an acceptable fit (where chi-square, *χ*^2^ = 3259.4, *p* = 0.000).

Furthermore, as shown in Table 6 below, the survey items demonstrated a high level of positive concordance (0.770). Also, the two-way mixed effect model revealed a good model for KH-11 with an average intraclass correlation of 0.80 at a 95% confidence interval (CI), where *p*-value = 0.000 (see Table 6 below).

### 4.5. Knowledge of Hydration among Foreign Students of Óbuda University, Hungary

The result captured in Figure 1 presents the pattern of knowledge of hydration among the study participants. The majority (66.3%) of the participants had adequate knowledge, while 24.4% had intermediate knowledge, and 9.3% had poor knowledge of hydration.

## 5. Discussion

We developed and conducted the psychometric analysis of a scale for the assessment of the Knowledge of Hydration (KH-11) among foreign students at Óbuda University, Hungary. The mean score was 55.2 ± 11.61, which indicates a relatively high level of knowledge among the study population. However, the distribution of the knowledge scores was skewed to the left, suggesting that some participants may have lower levels of knowledge about hydration. The reliability of the KH-11 tool was assessed using Cronbach’s alpha, which represents a measure of internal consistency. The overall reliability of the KH-11 tool was good, with a Cronbach’s alpha value of 0.80, indicating that the items in the survey were consistent in measuring the same construct of knowledge about the topic. Furthermore, the item-level consistency of the KH-11 tool was also evaluated using item Cronbach’s alpha values, which ranged from 0.75 to 0.81, which reveals an acceptable level of consistency among the items. This suggests that the items in the KH-11 were reliable in measuring the knowledge of the participants. In addition, Pearson’s correlation (R) within factors, which examined the independence of items within each factor, was positive and ranged from 0.02 to 0.74, which suggests that the items in the survey were sufficiently independent of each other. Additionally, this suggests that the items in the KH-11 tool were measuring different aspects of knowledge and were not overly correlated, supporting the construct validity of the survey. Hence, the scales of the KH-11 tool have recommended reliability and consistency. Conventionally, Cronbach’s alpha value > 70% is acceptable according to other researchers [28]. As compared to the Cronbach value revealed in this study, some validated scales had a lower value. For instance, the psychometric properties developed for a brief generic cancer knowledge for patients reported a lower Cronbach’s alpha value of 0.68 [29]. Although, another study on psychometric properties of the dementia knowledge assessment among home care workers in Taiwan reported Cronbach’s alpha slightly higher than the value we found in our study [30]. However, the value found in this study still falls within the acceptable threshold. Factors leading to a low Cronbach’s alpha value include fewer items in the scale and weakness of the set of items to measure the construct of interest [28].

The factor loadings for the KH-11 survey were generally remarkable, suggesting that the items in the survey effectively measured the underlying constructs of knowledge related to Total Water Intake (TWI) and its benefits, recommendations, and importance for staying hydrated. Specifically, all six (6) items in the KH-11 tool measuring the knowledge of the benefits of TWI loaded on factor 1 with factor loadings greater than 0.5 indicate a strong association between these items and the underlying construct of knowledge about the benefits of TWI. Similarly, the four (4) items measuring recommendations of TWI and its sources loaded on the second subscale with factor loadings greater than 0.9, thus indicating a very strong association between these items and the underlying construct of knowledge about recommendations and sources of TWI. Only one (1) item was loaded on the third subscale, measuring the importance of water intake for staying hydrated, with a factor loading greater than 0.9, which reveals a high association between this item and the underlying construct of knowledge about the importance of water intake for staying hydrated. Furthermore, the overall model fit was acceptable, with a chi-square value of 3259.4 and a *p*-value of 0.000, which demonstrates that the model adequately explained the observed data. The high concordance value of 0.770 among the items also indicates a high level of consistency in the responses, suggesting that the KH-11 tool is reliable in measuring knowledge about water intake. Furthermore, the two-way mixed effect model also revealed a good model for KH-11 with an average intraclass correlation of 0.80 at a 95% CI, with a *p*-value of 0.000, which signifies that the KH-11 survey has good reliability and consistency across different measurements.

The use of our study instrument has captured that the participants who have participated in this study had a good knowledge of hydration with a mean score of 55.2 ± 11.17 SD out of a total score of 72. This is relative to other studies carried out in a different context that have also reported a seemingly high level of knowledge on hydration [16]. However, this disparity is likely caused by the construct of the instrument used for measuring the referenced study. Our study developed and validated a scale for the assessment of knowledge in the light of the study context.

Factor analysis was used to examine the psychometric properties of the KH-11 tool. The result determined a three-factor structure for the scale. The items optimally loaded on the three subscales (Subscale 1. Knowledge of the benefits of Total Water Intake (TWI); Subscale 2. Recommendation of daily Total Water Intake (TWI) and source; and Subscale 3. Importance of water intake for staying hydrated). The results of the optimal factor loading with an acceptable score ranging from 0.5 to 0.9, the model fit, and the intraclass correlation imply that the construct validity for the KH-11 scale is acceptable. Thereby signifying this research stands out in its novelty in the development of an 11-item scale to measure the knowledge of hydration among students at Óbuda University, Hungary. Other studies have developed scales to measure outcomes that are different from the outcomes of interest in this study. Many of the studies also reported optimal results similar to this study [31]. The similarity in the findings could be linked to the use of the same methods of developing the scales and the method of analysis.

The mean score for the KH-11 survey was 55.2 ± 11.61 SD, out of a total possible score of 72. This suggests that, on average, the participants had a moderate level of knowledge about water intake as assessed by the KH-11 tool. Furthermore, the mean scores for the three subscales of the KH-11 were also reported. Subscale 1, which measures the importance of water intake for staying hydrated with only one item, had a mean score of 0.9 ± 0.07, indicating a high level of knowledge on this specific aspect. Subscale 2, which measures recommendations of Total Water Intake (TWI) with four items, had a mean score of 6.3 ± 3.60, indicating a moderate level of knowledge on this aspect. Subscale 3, which measures the knowledge of the benefits of TWI with six items, had a mean score of 47.9 ± 11.17, indicating a moderate level of knowledge on this aspect, respectively.

In furtherance, we categorized the knowledge score into three groups (<50% score = poor knowledge, 50–70% = intermediate knowledge, and 71–100% = adequate knowledge) to determine the level of knowledge of water intake among the students. The result showed that the majority (66.3%) of the students had good knowledge, while 24.4% had intermediate knowledge and 9.3% had poor knowledge of hydration. The rate of good knowledge reported in this study was quite higher compared to a similar study carried out in 2014 among adults in the United Kingdom, France, and Spain—the study from which we have adopted the survey questions [32]. Another study conducted in 2022 found a 51% rate of adequate knowledge of hydration among medical university students in UAE who correctly recognized the importance of water [33]. This prevalence is also lower compared to our findings. Similarly, Lee et al. reported a rate of 28% prevalence of adequate water intake (AWI) in 2016 [34]. The disparity in this finding could be due to the spatial distribution of the participants in the different studies. Also, the level of intervention toward improvement of well-being varied across countries [35]. In the same vein, there is a possible time difference in the findings; the pattern seems to increase over time. Factors such as context, population, and robust methodology of the current study play a key role in the disparities between this study and the other studies being compared. However, our study revealed a slightly higher prevalence of knowledge about hydration among students.

## 6. Conclusions

The KH-11 scales had a significant amount of acceptable reliability and consistency. Also, the items were optimally loaded on a three-factor structure for the scale, which implies construct validity. Hence, the use of this instrument captured that the participants had a good knowledge of hydration with a mean score of 55.2 ± 11.17 SD. The prevalence of an adequate level of knowledge of hydration was 66.3% in this study, which signifies a low knowledge level. This study suggests a need for further education and intervention to improve knowledge about hydration among foreign students, which can have important implications for their overall health and well-being. Overall, the results of the study indicate that the KH-11 is a reliable and valid measure of knowledge about the topic under investigation, with good overall reliability and acceptable item-level consistency. However, the skewed distribution of knowledge scores suggests that some participants may have lower levels of knowledge, which may warrant further investigation or intervention to improve knowledge in those individuals. Therefore, this research has implemented a second phase to address these concerns through the Hydration Campaign project at Óbuda University, which aims to improve the student’s knowledge of hydration. The second phase of the project commenced in April 2023 till date.

## 7. Recommendation

We recommend the use of this scale for future research in the context in which the scale was developed. Although this study revealed an appreciable level of knowledge of hydration, intervention should be made to improve knowledge to 100% among the students of John von Neumann Faculty of Informatics at Óbuda University, Hungary. Hence, we have considered the need to begin a campaign to include the entire students (foreign and local Hungarian) of the university: Hydration Campaign at Óbuda University.

## 8. Study Limitations and Future Work

Due to the unavailability of adequate resources, we could only reach out to a fraction of the students of Óbuda University during the survey distribution. Furthermore, the use of hydration biomarkers was not implemented to diagnose hydration status in this study. However, this does not erode the significance of this paper, as probabilistic samples were drawn from the target population. In order to correct the shortcomings of the unavailability of sufficient data for this study, we are currently organizing a hydration campaign among the (foreign and local) PhD students of Óbuda University to create more awareness about hydration, assess the knowledge of the students about hydration, and also assess their knowledge using the second survey to estimate if there is an improvement on the knowledge about hydration after taking the first survey.

## Figures and Tables

**Figure 1 healthcare-12-01152-f001:**
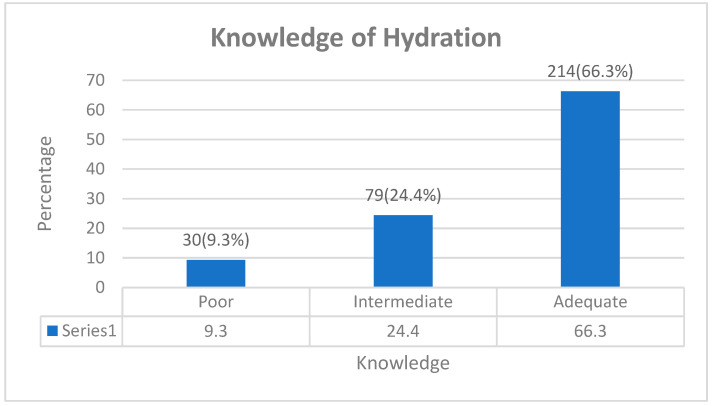
Knowledge of Hydration among Foreign Students of Óbuda University, Hungary.

**Table 1 healthcare-12-01152-t001:** Knowledge of Hydration 11-item scale (KH-11) Survey Questions.

SN	Questions—KH-11	Options and Scores
1	Do you think drinking water is important to help stay properly hydrated?	Yes = 1 No = 0
2	What do you think is the recommended daily Total Water Intake (TWI) (in Liters per day)? Please, answer for both males and females.	Male = 2.5 = 2 Female = 2.0 = 2 Else = 0
3	According to the European Food Safety Authority (EFSA) or World Health Organization (WHO), what do you think is the recommended daily Total Water Intake (TWI) (in Liters per day)? Please, answer for both males and females.	Male = 2.5 = 2 Female = 2.0 = 2 Else = 0
4	The daily recommended water intake has to come from which of the following?	[Food moisture only] = 0, [Beverages of all kinds only] = 0, [Water only] = 0 [All of the Above = 4] = 1
5	According to the European Food Safety Authority (EFSA) or World Health Organization (WHO), the daily recommended Total Water Intake (TWI) has to come from which of the following?	[Food moisture only] = 0, [Beverages of all kinds only] = 0, [Water only] = 0 [All of the Above = 4] = 1
6	Please carefully read each of the following statements about hydration during different moments in your life. Decide by using a scale of 1–10 (note: the scale increases from 1 to 10) Each of the 6 items has a score ranging from 1–10 summed up to 60 possible score	a. In an adult, drinking a lot of water is a good way to cleanse the body (1–10) b. At school and work, Adequate Water Intake (AWI) is important for proper brain performance and productivity (1–10); c. During the day everyone should drink at least 2.0 L of water (1–10) d. Staying hydrated (water intake) can be influenced by some factors (such as Exercise Intensity and Duration, Temperature and Humidity, etc. (1–10) e. Due to age, the sensation of thirst can be reduced (1–10) f. Increased water intake contributes to living a healthy life (1–10)
11 items, the total possible score is 72. High scores imply good knowledge		

**Table 2 healthcare-12-01152-t002:** Demographic Characteristics of the Participants.

Variables	Frequency (n = 323)	Percent (%)
**Age Group**		
Between 18 and 24	169	51.7
Between 25 and 35	156	48.3
**Gender**		
Female	141	43.6
Male	182	56.4
**Continent of Origin**		
Africa	104	32.1
Asia	101	31.3
Europe	118	36.5
**Place of Residence**		
Hungary	323	100.0

**Table 3 healthcare-12-01152-t003:** The Distribution and Characteristics of the KH-11.

Mean (SD)	No of Items	Cronbach’s Alpha	Shapiro–Wilk	*p*-Value
55.2 (11.61)	11	0.80	0.89	0.000

**Table 4 healthcare-12-01152-t004:** Knowledge about Hydration.

Subscales	Mean (SD)	Total Possible Score
Subscale 1: Importance of water intake for staying hydrated [one item].	0.9 (0.07)	1
Subscale 2: Recommendation of Total Water Intake (TWI) and source (Personal recommendation and recommendation according to the European Food Safety (EFSA) and the World Health Organization (WHO) [four items].	6.3 (3.60)	11
Subscale 3: Knowledge of the benefits of Total Water Intake (TWI) [six items].	47.9 (11.17)	60
Total	55.2 (11.61)	72

**Table 5 healthcare-12-01152-t005:** Factor Analysis Results for KH-11.

Items	Factors			Chi-Square	*p*-Value
	1	2	3	3259.4	0.000
Do you think drinking water is important to help stay properly hydrated?			0.97		
What do you think is the recommended daily Total Water Intake (TWI) (in Liters per day)?		0.91			
According to the European Food Safety Authority (EFSA) or World Health Organization (WHO), what do you think is the recommended daily Total Water Intake (TWI) (in Liters per day)?		0.91			
The daily recommended Total Water Intake (TWI) has to come from which of the following?		0.91			
According to the European Food Safety Authority (EFSA) or World Health Organization (WHO), the daily recommended Total Water Intake (TWI) has to come from which of the following?		0.91			
Please carefully read each of the following statements about hydration during different moments in your life					
In an adult, drinking a lot of water is a good way to cleanse the body.	0.86				
At school and work, adequate hydration is important for proper brain performance and productivity.	0.89				
During the day everyone should drink at least 2.0 L of water.	0.86				
Staying hydrated can be influenced by some factors (such as Exercise Intensity and Duration, Temperature and Humidity, etc.).	0.84				
Due to age, the sensation of thirst can be reduced.	0.51				
Increased water intake contributes to living a healthy life.	0.87				
Kendal Tau coefficient of concordance					0.77

**Rotated Component Matrix.** Factor 1: Knowledge of the benefits of Adequate Water Intake (AWI). Factor 2: Recommendation of Adequate Water Intake (AWI) and source. Factor 3: Importance of water intake for staying hydrated.

**Table 6 healthcare-12-01152-t006:** Two-way Mixed Effect Model for KH-11.

	Intraclass Correlation	95% CI		F Test	
		Lower Bound	Upper Bound		*p*-value
Average Measures	0.80	0.767	0.827	4.952	0.000

## Data Availability

The hydration and dehydration data collected during this project and processed for this research publication are available upon request. Data is contained within the article.

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
