# Peer review of "Psychometric Properties of the Knowledge of Hydration among Foreign Students of Óbuda University, Hungary"

_healthcare, 2024, doi:10.3390/healthcare12111152_

Round 1

Reviewer 1 Report

Comments and Suggestions for Authors 1-When measuring Psychometric Properties, there are many factors that affect  including nutrition (which includes water, of course), biochemical analysis such as liverand kidney function, mineral, endocrinology analysis, etc ), environmental factors, and lifestyle, family history, beverages favorite among students, appetite is matter. We should pay more attention to these factors. The authour should give more attention towards those factor in introduction and discussion.

2-The methodology need more attention. Comments on the Quality of English Language

The manuscript should be accepted after check for plagiarism,  AI checking and proofreading , if any .

Author Response

Reviewer 1

Comments and Suggestions for Authors

1-When measuring Psychometric Properties, there are many factors that affect including nutrition (which includes water, of course), biochemical analysis such as liver and kidney function, mineral, endocrinology analysis, etc.), environmental factors, and lifestyle, family history, beverages favorite among students, appetite is matter. We should pay more attention to these factors. The author should give more attention towards those factor in introduction and discussion.

Note to Reviewer 1 from Author

Thank you for your comprehensive review of this manuscript; your input is greatly appreciated as it enhances the quality of the document. Please note that this paper is not focusing on the clinical evaluation of hydration but on the knowledge among students (not medical practitioners/students). Hence, it will be non-systematic/scientific to measure the knowledge of hydration biomarkers among this population. However, it's important to note that in this study, we only evaluated the reliability of the questionnaire using a reliability test, and we reported the Cronbach's alpha. The psychometric properties were assessed for knowledge in this research. Thus, we are not able to perform the following because this is not an experimental study, but a collection of self-reported data only to measure the participants' level of knowledge as regards hydration.

  • Biochemical analysis such as liver and kidney function, mineral, endocrinology analysis, etc), environmental factors.
  • Environmental factors.
  •  

As regards other factors mentioned below, we have another manuscript currently ongoing to discuss these factors. We have not included all the factors here because the aim of this study is to validate the tool.

  • Beverages favourite (type and in liters).
  • Appetite.

Considering the above, we have also mentioned it in the study limitation section of this study. Please, see Section 8. Study Limitations and Future Work.

Due to the unavailability of adequate resources, we could only reach out to a fraction of the students of Óbuda University during the survey distribution. Furthermore, the use of hydration biomarkers was not implemented to diagnose hydration status in this study. However, this does not erode the significance of this paper, as probabilistic samples were drawn from the target population.

2-The methodology needs more attention.

The methodology has been revised.

Comments on the Quality of English Language

The manuscript should be accepted after checking for plagiarism,  AI checking and proofreading , if any .

The quality of English has been checked.

Reviewer 2 Report

Comments and Suggestions for Authors

The manuscript was very interesting and valuable and I enjoyed reading it

I offer some suggestions to make the study more clear

Provide statistics of average water consumption among students

What method did you use to determine the sample size?

   Explain the questionnaire design method. Have you used the opinions of experts for the validity of the content?

  Who was the decision-making team for preparing the questionnaire? What did they specialize in?

  Explain the method of sampling better, was it correct?

  Was written consent obtained from the participants to participate in the study?

Author Response

Reviewer 2

The manuscript was very interesting and valuable and I enjoyed reading it

I offer some suggestions to make the study more clear.

Provide statistics of average water consumption among students

Thank you for your comprehensive review of this manuscript; your input is greatly appreciated as it enhances the quality of the document. This is included in an ongoing paper and also we have an existing paper which includes this aspect. This current paper only validates the scales. Hence, we consider this aspect not relevant to this paper. This study has accessed knowledge.

M. Omone, M. Kozlovszky, T. Ferenci, and I. G. Inalegwu, “Hydration Assessment Among Foreign University Students,” in 2019 IEEE 19th International Symposium on Computational Intelligence and Informatics and 7th IEEE International Conference on Recent Achievements in Mechatronics, Automation, Computer Sciences and Robotics (CINTI-MACRo), Nov. 2019, pp. 000161–000168. doi: 10.1109/CINTI-MACRo49179.2019.9105318.

What method did you use to determine the sample size?

The sample size method is explained. Please, see Section 3.2.

   Explain the questionnaire design method. Have you used the opinions of experts for the validity of the content?

The questionnaire design method is explained. Please, see Section 3.1.

  Who was the decision-making team for preparing the questionnaire? What did they specialize in?

The decision-making team includes the following;

  • Melvin Omone Ogbolu – Bioinformatics (Biostatistics and Biomedical Engineering)
  • Olanrewaju D. Eniade – Epidemiology and Medical Statistics
  • Zrubka Zsombor János – Health Economics, Oncology, and Biotechnology
  • Miklós Kozlovszky – Bioinformatics (Telemedicine and Biomedical Engineering)

Explain the method of sampling better, was it correct?

The method of sampling has been explained. Please, see Section 3.2.

  Was written consent obtained from the participants to participate in the study?

The data used for this study is self-reported data from a survey we developed via the Surveyplanet website. Before the participants were administered the survey, there was a consent survey which allowed the participants to decide if they were willing to participate in the entire project (seminars and surveys) if invited or not. The first page of the questionnaire contained the informed consent which consisted of information about the study, voluntary participation, and the ability to withdraw consent at any point. Each survey link sent via email or posted on the University’s website or Facebook page also had a statement asking that the participants proceed to answer questions if only they want to participate in the study. However, all participants remain completely anonymous.

Reviewer 3 Report

Comments and Suggestions for Authors

·         Abstract needs to be revised and should cover the methodology section in brief.

·         Motivation is not clear in the scenario of water deficiency problem. it should include.

·         The methodology section did not cover all aspects of data collection. include all aspects and condition, data type and design in clarity.

·         Line 343 to 396 are not clear. As it is MDPI format copied but no content related to manuscript is available, revise all last lines.

·         Include recent references of food and Hydration.

·         Table 2. Demographic Characteristics of the Participants, need to cover job , occupation and physical activity.

·         The Questions – KH-11 questionnaire is not sufficient to decide the comments and conclusion based on online feedback. It needs to revise.

Comments on the Quality of English Language

Dear Author, 

 Extensive editing of English language and connection is required in this manuscript.

Author Response

Note to Reviewer 3 from Author

  • The abstract needs to be revised and should cover the methodology section in brief.

Thank you for your comprehensive review of this manuscript; your input is greatly appreciated as it enhances the quality of the document. The abstract has been revised. Summary of the results and methodology added.

  • Motivation is not clear in the scenario of the water deficiency problem. it should include.

The motivation has been revised.

  • The methodology section did not cover all aspects of data collection. include all aspects and conditions, data type and design in clarity.

Methodology explained. Please, see Section 3.1 – 3-7.

  • Lines 343 to 396 are not clear. As it is MDPI format copied but no content related to the manuscript is available, revise all last lines.

Lines 343 – 396 contain the backmatter of this article and are all relevant to the study.

  • Include recent references to food and Hydration.

Recent references have been added.

  • Table 2. Demographic Characteristics of the Participants, need to cover job, occupation and physical activity.

The occupation was not included in the instrument (the study population are students).

All participants were students when this study was conducted. Physical activity data were not collected for this study which has been stated as one of the limitations of the study.

Considering the above, we have also mentioned it in the study limitation section of this study. Please, see Section 8. Study Limitations and Future Work

Due to the unavailability of adequate resources, we could only reach out to a fraction of the students of Óbuda University during the survey distribution. Furthermore, the use of hydration biomarkers was not implemented to diagnose hydration status in this study. However, this does not erode the significance of this paper, as probabilistic samples were drawn from the target population.

  • The Questions – KH-11 questionnaire is not sufficient to decide the comments and conclusion based on online feedback. It needs to be revised.

Thank you for your comprehensive review of this manuscript; your input is greatly appreciated as it enhances the quality of the document. Please note that this paper is not focusing on the clinical evaluation of hydration but on the knowledge among students (not medical practitioners/students). Hence, it will be non-systematic/scientific to measure the knowledge of hydration biomarkers among this population. However, it's important to note that in this study, we only evaluated the reliability of the questionnaire using a reliability test, and we reported the Cronbach's alpha. The psychometric properties were assessed for knowledge in this research.

Round 2

Reviewer 3 Report

Comments and Suggestions for Authors

Dear Author, 

The suggested comments address and revised manuscript in good shape.

Comments on the Quality of English Language

Extensive English is required i whole manuscript